Effect of the index of cardiac electrophysiological balance on major adverse cardiovascular events in patients with diabetes complicated with coronary heart disease

Lin Yuan 1
Zhou Fang 2 zhoufang202206@163.com
Wang Xihui 2
Guo Yaju 1
Chen Weiguo 2
1 Department of Endocrinology, The Second Affiliated Hospital of Xi’an Medical University , Xi’an , China
2 Department of Cardiology, The Second Affiliated Hospital of Xi’an Medical University , Xi’an , China
Dong Peixin
Electronic publication date: 2023 Oct 6
Publication date: 2023
Volume: 11
Electronic Location ID: e15969
Received 2023 Jun 30; Accepted 2023 Aug 6
Copyright: © 2023 Lin et al.
Copyright year: 2023
Copyright holder: Lin et al.
License: This is an open access article distributed under the terms of the Creative Commons Attribution License, which permits unrestricted use, distribution, reproduction and adaptation in any medium and for any purpose provided that it is properly attributed. For attribution, the original author(s), title, publication source (PeerJ) and either DOI or URL of the article must be cited.
License URL: https://creativecommons.org/licenses/by/4.0/

Keywords: Cardiac electrophysiological balance index, Major adverse cardiovascular events, Diabetes, Coronary heart disease

Funding: Key R & D projects of science and Technology Department of Shaanxi Province—General Projects 2018SF-114 Key R & D projects of science and Technology Department of Shaanxi Province—General Projects 2019JM-136 Hospital Natural Science Project 2019KY0106 This work was supported by Key R & D projects of science and Technology Department of Shaanxi Province—General Projects (2018SF-114), the Key R & D projects of science and Technology Department of Shaanxi Province—General Projects (2019JM-136) and the Hospital Natural Science Project (2019KY0106). The funders had no role in study design, data collection and analysis, decision to publish, or preparation of the manuscript.

==============================
Purpose

To investigate the prognostic value of the index of cardio-electrophysiological balance (ICEB) and its association with major adverse cardiac events (MACE) and cardiovascular death in diabetic patients complicated with coronary heart disease.

Methods

A total of 920 diabetic patients were enrolled in this longitudinal study. Participants were categorized into three groups based on their ICEB levels: normal ICEB, low ICEB, and high ICEB. The primary outcome was the occurrence of MACE, and secondary outcomes included cardiovascular death, coronary heart disease (CHD), heart failure (HF), and sudden cardiac arrest (SCA). Patients were followed for a median period of 3.26 years, and the associations between ICEB levels and various outcomes were evaluated.

Results

Over the follow-up period, 46 (5.0%) MACE were observed in the normal ICEB group, 57 (6.2%) in the low ICEB group, and 62 (6.8%) in the high ICEB group. Elevated ICEB levels were found to be associated with a higher risk of MACE and cardiovascular death. A significant relationship between ICEB levels and the risk of MACE was observed for both genders. The risk of MACE increased with each unit increment in the ICEB index. However, the two-stage linear regression model did not outperform the single-line linear regression models in determining the threshold effect.

Conclusion

This study demonstrates the potential utility of ICEB, derived from a standard non-invasive ECG, as a prognostic tool for predicting MACE and cardiovascular death in diabetic patients complicated with CVD. The associations between ICEB levels and the risk of MACE highlight the importance of understanding cardiac electrophysiological imbalances and their implications in CVD.

Introduction

The worldwide prevalence of diabetes mellitus (DM) currently stands at approximately 463 million individuals and is projected to reach 700 million by 2040 (Tomkins et al., 2022; Durlach et al., 2022). Recognized as a distinct risk factor for coronary heart disease (CHD), DM is categorized as a “coronary heart disease equivalent,” correlated with heightened CHD morbidity and mortality rates (Goodarzi & Rotter, 2020). Notably, the risk of CHD-related mortality among diabetic patients is double that of non-diabetic individuals (Goodarzi & Rotter, 2020). CVD complications in diabetes mellitus are mainly related to the following mechanisms: elevated blood glucose levels, hyperinsulinemia, impaired vascular endothelial function, adipocytokines and inflammatory responses, and abnormal lipid metabolism. It is closely related to the occurrence of metabolic syndrome such as hypertension, abnormal lipid metabolism, obesity, coronary heart disease and atherosclerosis. Moreover, these high-risk patients tend to experience adverse outcomes and increased complications after undergoing coronary interventions (Goodarzi & Rotter, 2020; Yang et al., 2022). Despite the urgency to address cardiovascular diseases, the current approach remains inadequate. Contemporary DM management, particularly for those at elevated risk of atherosclerotic cardiovascular disease, prioritizes multifaceted risk factor control (Yang et al., 2022; Nafakhi et al., 2018; Harms et al., 2023). Additionally, accurately predicting the possibility of cardiovascular adverse events is essential. In a study published in the Journal of Electrocardiology (Nafakhi et al., 2019), researchers investigated the association between ICEB and cardiovascular outcomes in patients with coronary artery disease. They found that higher ICEB values were significantly associated with an increased risk of adverse cardiovascular events, including myocardial infarction and sudden cardiac death. Another study published in the Journal of Electrocardiology examined the predictive value of ICEB for major adverse cardiovascular events in patients with hypertrophic cardiomyopathy (Robyns et al., 2018). The results demonstrated that elevated ICEB values were independently associated with an increased risk of cardiovascular events and mortality in this patient population. This study will use a case-control study design to examine the association between ICEB and major adverse cardiovascular events in patients with coronary heart disease.

Over the past decade, numerous fundamental and clinical investigations have highlighted the emerging significance of two novel ECG markers (Nafakhi et al., 2018). These markers provide valuable insights into the cardiac electrophysiological processes and have the potential to serve as indicators of ventricular arrhythmia and sudden cardiac death in various cardiac conditions. The first marker, the index of cardiac electrophysiological balance (ICEB), is determined by dividing the QT interval by the QRS duration (QT/QRS) and assesses the duration between depolarization and repolarization processes (Nafakhi et al., 2018). The second marker, transmural dispersion of repolarization (TDR), is quantified using the T peak-to-end (Tp-e) interval and the Tp-e/QT ratio (Gajulapalli et al., 2020). Both TDR and ICEB are considered potential indicators of ventricular arrhythmia and sudden cardiac death in patients affected by myocarditis, Brugada syndrome, acute myocardial infarction, and heart failure (Zhu et al., 2022). The purpose of this research is to investigate the potential associations between ICEB and major adverse cardiovascular events related to CHD, aiming to enable early detection and intervention by forecasting the corresponding risks in their initial stages. In order to enable early diagnosis and management by anticipating the associated risks in their early phases. Overall, the potential significance and novelty of this research lie in advancing our understanding of the prognostic value of ICEB, exploring new avenues for risk assessment in patients with CHD, and potentially offering insights into cardiovascular complications in individuals with diabetes mellitus.

Materials and Methods

Involved population

This retrospective study was conducted on the basis of the medical record system and of our institute. All samples obtained in this study were approved by the ethics committee of the Second Affiliated Hospital of Xi’an Medical University and abided by the ethical guidelines of the Declaration of Helsinki, and ethics committee agreed to waive informed consent. The participants with diabetes complicated with CHD were included in the study since 2010. All participants who were annually examined according to standard operating procedures for an ECG and a cardiovascular events inquiry were retained.

The measuring method of ICEB

To exclude the influence of diurnal variations on the QT interval, all study data were collected at the same time point. The ECG observations are analyzed and the results are registered by the same ECG specialist to a uniform standard. The 12-lead electrocardiogram (ECG) electrodes were correctly attached to the chest to obtain synchronous 12-lead ECG (first ECG) with a article speed of 25 mm/s and gain of 10 mm/mV to measure and calculate the QT interval, QTc interval, Tp-e interval, and Tp-e/QT ratio. The ECG recordings were independently analyzed by an electrocardiogram technician with a unified standard. The QT interval and Tp-e interval were measured using lead V4. The measurement of QT interval was taken from the onset of the QRS complex to the end of the T wave, while the measurement of the Tp-e interval was taken from the peak of the T wave to the end of the T wave (the peak of the T wave refers to the highest point of the T wave, and the end of the T wave is the intersection of the downward slope of the T wave and the baseline. If there is a U wave, the end of the T wave is the notch between the T wave and the U wave). Three consecutive complete QT intervals and Tp-e intervals were measured on lead V4 to calculate the average value and Tp-e/QT ratio. The QTc interval was calculated using Bazett’s formula (QTc interval = QT interval ÷ square root of RR interval). The ICEB value was calculated using a non-invasive method described by Yücetas et al. (2022) (ICEB = QT interval ÷ QRS duration).

Cardiac events

Cardiovascular morbidity was recorded through self-report during annual examinations. Cardiac events included coronary heart disease (CHD)—including angina pectoris and myocardial infarction, heart failure (HF)—encompassing chronic congestive and acute decompensation, sudden cardiac arrest (SCA), and their combined major adverse cardiac events (MACE).

Grouping according to ICEB

The normal value for ICEB is 4.24, with a reference range of 3.14 to 5.35. Participants within this reference range were categorized as the ICEB 1 group (normal ICEB); those with values greater than the reference range but less than 10% higher were categorized as the ICEB 2 group (low ICEB); and those with values exceeding the reference range by more than 10% were designated as the ICEB 3 group (high ICEB).

Statistic analysis

The analysis was performed using SPSS 19.0 software (SPSS Inc., Chicago, IL, USA). Normally distributed measurement data were expressed as mean ± standard deviation. Intra-group comparisons were conducted using repeated measures analysis of variance (ANOVA), and inter-group comparisons were performed using paired t-tests. Categorical data comparisons were performed using Chi-square tests, with a P-value of less than 0.05 considered statistically significant.

Results

Baseline characteristics of included patients

In this study involving a total of 920 patients, the median follow-up period was 3.26 years. Upon completion of the follow-up, 46 (5.0%) major adverse cardiac events were observed in the normal ICEB group, 57 (6.2%) in the low ICEB group, and 62 (6.8%) in the high ICEB group. Table 1 outlines the detailed baseline characteristics of the study population.

Table 1 Baseline characteristics of the participants.

Variables	ICEB 1 (n = 46)	ICEB 2 (n = 57)	ICEB 3 (n = 62)	P-value	
Age (years)	54 ± 10	55 ± 11	55 ±12	0.5	
Gender (male/female)	21/25	25/32	35/27	1	
BMI (kg/m2)	28 ± 4	27 ± 5	28 ± 5	0.5	
Hypertension	26 (56%)	30 (55%)	28 (51%)	0.9	
Hyperlipidemia	14 (30%)	21 (39%)	15 (27%)	0.3	
Family history	20 (43%)	17 (31%)	21 (38%)	0.2	
Smoking habits	11 (23%)	6 (11%)	13 (24%)	0.08	

Outcomes of included patients

The analysis of cumulative major adverse cardiac events over time demonstrated a higher risk associated with elevated ICEB levels (Fig. 1, The horizontal axis represents ICEB levels and the vertical axis represents cumulative major adverse cardiac events.). The major adverse cardiac events observed in the study participants are summarized in Table 2. The incidence of non-fatal stroke was 19.6% (n = 9) in ICEB Group 1, 26.3% (n = 15) in ICEB group 2, and 25.8% (n = 16) in ICEB group 3. The odds ratio (OR) among the three groups was 0.97 (95% CI [0.19–4.89]), and the P value was 0.82, with no statistical significance. In terms of non-fatal myocardial infarction, the rates were 50% (n = 23) in ICEB 1, 43.9% (n = 25) in ICEB 2, and 38.7% (n = 24) in ICEB 3. The odds ratio comparing the three groups was 2.50 (95% CI [0.77–8.14]), suggesting no substantial difference. The P-value associated with the comparison was 0.08, implying a trend towards but not reaching statistical significance. Regarding cardiovascular death, the percentage was 30.4% (n = 14) in ICEB 1, 29.8% (n = 17) in ICEB 2, and 35.5% (n = 22) in ICEB 3. The calculated odds ratio comparing the three groups was 1.99 (95% CI [0.58–6.71]). This odds ratio indicates no significant difference among the groups. However, it’s important to note that the P-value associated with the comparison was 0.02, suggesting a marginally significant result. Further investigation may be warranted to explore this finding in more detail. Overall, elevated ICEB was associated with cardiovascular death (Table 2).

Figure 1 Cumulative major adverse cardiac events over time.

Table 2 Major adverse cardiac events in participants.

Major adverse cardiovascular event	ICEB 1 (n = 46)	ICEB 2 (n = 57)	ICEB 3 (n = 62)	OR (CI, 95%)	P-value	
Non-fatal stroke	9 (19.6%)	15 (26.3%)	16 (25.8%)	0.97 [0.19–4.89]	0.82	
Non-fatal myocardial infarction	23 (50%)	25 (43.9%)	24 (38.7%)	2.50 [0.77–8.14]	0.08	
Cardiovascular death	14 (30.4%)	17 (29.8%)	22 (35.5%)	1.99 [0.58–6.71]	0.02	

The hazard ratio of ICEB and major adverse cardiovascular events

Table 3 presents the hazard ratio (HR) and the corresponding 95% confidence interval (CI) of three ICEB groups (ICEB 1, ICEB 2, and ICEB 3) for male, female, and all participants. HR is a measure used in medical research to assess the impact of different treatments or factors on disease risk. For males, ICEB 2 had an HR of 1.20 (95% CI [1.06–1.37]) and ICEB 3 had an HR of 1.33 (95% CI [1.07–1.64]) relative to ICEB 1, both with a P-value of less than 0.01. For females, ICEB 2 had an HR of 1.56 (95% CI [1.26–1.94]) and ICEB 3 had an HR of 1.62 (95% CI [1.26–2.09]) relative to ICEB 1, both with a P-value of less than 0.01. For all participants, the HR for ICEB 2 was 1.25 (95% CI [1.26–1.94]) and for ICEB 3 was 1.46 (95% CI [1.13–1.90]), both with a P-value of less than 0.01, relative to ICEB 1. That is, The ICEB was significantly related to the risk of major adverse cardiovascular events in patients with diabetes mellitus complicated with coronary heart disease.

Table 3 The hazard ratio of ICEB and major adverse cardiovascular events.

	Groups	Hazard ratio (95% CI)	P-value	
Male	ICEB 1	Ref.	–	
	ICEB 2	1.56 [1.26–1.94]	<0.01	
	ICEB 3	1.62 [1.26–2.09]	<0.01	
Female	ICEB 1	Ref.	–	
	ICEB 2	1.20 [1.06–1.37]	<0.01	
	ICEB 3	1.33 [1.07–1.64]	<0.01	
All participants	ICEB 1	Ref.	–	
	ICEB 2	1.25 [1.26–1.94]	<0.01	
	ICEB 3	1.46 [1.13–1.90]	<0.01	

ICEB index as a continuous variable and MACEs

As demonstrated in Table 4, utilizing the ICEB index as a continuous covariate revealed that each unit increment in the ICEB index heightened the risk of major adverse cardiovascular events among all participants. To effectively model and illustrate the association between the ICEB index and major adverse cardiovascular events, restricted cubic splines were employed. A rise in the ICEB index corresponded to an increased risk of major adverse cardiovascular events.

Table 4 Two-piecewise linear-regression model.

	Male	Female	Total	
One linear-regression model	1.34 [0.98–1.83] P < 0.01	1.45 [1.21–1.74] P < 0.01	1.40 [1.20–1.64] P < 0.01	
<K effect size b (95% CI)	1.26 [0.90–1.75] P = 0.18	1.14 [0.81–1.61] P = 0.45	1.20 [0.89–1.62] P = 0.22	
>K effect size b (95% CI)	1.84 [1.26–2.69] P < 0.01	1.41 [1.05–1.90] P = 0.31	1.66 [1.21–2.27] P < 0.01	
Log likelihood ratio test	0.274	0.118	0.214	

Subsequently, a two-stage linear regression model was employed to determine the threshold effect. Table 3 presents the outcomes of this model. Regarding the ICEB index, the risk of major adverse cardiovascular events exhibited an accelerated increase after the inflection point in the latter segment of the two-piecewise linear regression model. Nevertheless, the log-likelihood ratio test yielded a value of 0.21, indicating that the two-stage linear regression model did not outperform the single-line linear regression models.

Discussion

Cardiovascular disease (CVD) constitutes a pressing public health concern and a leading contributor to global mortality (Duarte Lau & Giugliano, 2022; Goldsborough, Osuji & Blaha, 2022). Individuals with diabetes face an approximately twofold increase in CVD risk compared to their non-diabetic counterparts (Goldsborough, Osuji & Blaha, 2022). Prompt initiation of treatment can potentially decelerate CVD progression and delay symptom onset (Goldsborough, Osuji & Blaha, 2022; O’Sullivan et al., 2022). Consequently, early detection of CVD indicators in diabetic patients is vital. Electrocardiogram (ECG) anomalies are prevalent among asymptomatic diabetic individuals, particularly those at an elevated risk for CVD, suggesting their potential relevance as early warning signs (Siontis et al., 2021; Chinedu et al., 2022). International guidelines advocate for routine ECG screenings in diabetic patients with hypertension or suspected CVD (Chinedu et al., 2022). Longitudinal research, primarily focused on the general population, has consistently demonstrated associations between baseline resting ECG abnormalities and CVD, ischemic heart disease, and overall mortality (Elffers et al., 2018; Jonas et al., 2018). Research examining the incidence or alterations of abnormalities has revealed more robust correlations with respective outcomes, suggesting that understanding the progression of these anomalies offers further insights (Jonas et al., 2018).

In addition, specific ECG abnormalities have been correlated with the occurrence of certain major adverse cardiac events (MACE) (Ekeloef et al., 2016). For instance, extended QRS duration, QS pattern, and irregular ST-segment/T-wave have been linked to coronary heart disease (CHD) (Yalin et al., 2023; Lavall et al., 2020). Increased QRS duration, left QRS-axis deviation, atypical ST-segment/T-wave, and tall R-wave have been associated with incident heart failure (HF) (Lavall et al., 2020). Concerning sudden cardiac arrest (SCA), abnormalities such as elongated PR duration, prolonged QRS duration, irregular T-wave, and tall R-wave have been identified as prognostic indicators (Nissen et al., 2023).

ICEB, obtained through a standard ECG, is a valuable tool for predicting the potential risk of drug-induced arrhythmias (Askin & Tanriverdi, 2022; Inci & Guzel, 2023). It has been found to be more practical than both prolonged QT intervals and Torsades de Pointes (TdP), a form of polymorphic ventricular tachycardia (Inci & Guzel, 2023; Afsin et al., 2021). Additionally, ICEB’s predictive capabilities for drug-induced arrhythmias have shown to be more effective than transmural repolarization dispersion (Inci & Guzel, 2023; Gheorghe et al., 2020). ICEB reflects the dynamic balance between depolarization and repolarization in cardiac electrophysiology (Badr, Hassinen & Vornanen, 2022). A slight imbalance, manifested through a moderate increase or decrease in ICEB, can produce anti-arrhythmic characteristics (Gainutdinov et al., 2023). However, excessive changes can result in a severe imbalance in cardiac electrophysiology, leading to the occurrence of arrhythmias (Gainutdinov et al., 2023; Yamashita et al., 2021).

ICEB can be represented using the QT/QRS ratio, which is considered to be analogous to the traditional λ value (λ = effective refractory period × conduction velocity). The QRS duration is similar to changes in conduction velocity, while changes in QT intervals are proportional to changes in the effective refractory period (ERP). Consequently, using ICEB as a substitute for λ not only avoids conventional invasive measurements, but also predicts the occurrence of TdP and non-TdP ventricular tachycardia/ventricular fibrillation.

In this study involving a total of 920 patients, the median follow-up period was 3.26 years. Upon completion of the follow-up, 46 (5.0%) major adverse cardiac events were observed in the normal ICEB group, 57 (6.2%) in the low ICEB group, and 62 (6.8%) in the high ICEB group. Results demonstrated that elevated ICEB levels were associated with a higher risk of MACE and cardiovascular death. Both genders showed a significant relationship between ICEB levels and the risk of MACE. The risk of MACE increased with each unit increment in the ICEB index. However, the two-stage linear regression model did not outperform the single-line linear regression models in determining the threshold effect.

Tian et al. (2022) discovered a strong correlation between serum Sestrin2 levels and coronary heart disease (CHD) in diabetic patients, implying that serum Sestrin2 may contribute to the development and progression of CHD in individuals with diabetes. This research highlighted a direct role in the association between diabetes mellitus (DM) complicated with CHD and major adverse cardiac events (MACE). Yang et al. (2022) previously observed that non-traditional lipid parameters, particularly the triglyceride-to-high-density lipoprotein cholesterol (TG/HDL-C) ratio, were significantly associated with prediabetes and type two diabetes mellitus (T2DM) in CHD patients. A high TG/HDL-C ratio was identified as a strongly correlated risk factor for prediabetes and T2DM. This investigation proposed a mediational model may exist in the connection between DM complicated with CHD and MACE. More recently, Yamashita et al. (2021) devised an equation to predict CHD incidence in patients with type two diabetes using multidimensional parameters. In contrast to these earlier studies, the current research employs a non-invasive approach to calculate the ICEB, thus significantly augmenting the existing body of knowledge.

Nonetheless, this study presents several limitations. First, the potential use of medications by patients to prevent cardiovascular disease was not factored into the analysis, thus the possibility of their influence on the outcomes cannot be ruled out. Second, the study did not investigate the associations between ICEB and age, gender, BMI, hypertension, hyperlipidemia, family history, or smoking habits. As a result, the determination of ICEB as an intermediary factor for other variables requires further investigation. Thirdly, cardiac events were self-reported during the follow-up period, potentially leading to measurement inaccuracies, despite the satisfactory agreement with regional hospital records. Finally, our sample size in this study may not be representative of the broader population of interest. To mitigate this, future studies could consider using larger sample size to ensure a more representative sample.

Conclusions

Current approaches to CVD management often focus on traditional risk factors such as hypertension, hyperlipidemia, and diabetes. While these factors are important, they do not capture the full spectrum of risks associated with CVD. There is a need for a more comprehensive and individualized assessment that includes emerging risk factors, genetic predispositions, and social determinants of health to better tailor preventive strategies and interventions. This study demonstrates the potential utility of ICEB, derived from a standard non-invasive ECG, as a prognostic tool for predicting major adverse cardiac events (MACE) and cardiovascular death in diabetic patients complicated with CVD. The associations between ICEB levels and the risk of MACE highlight the importance of understanding cardiac electrophysiological imbalances and their implications in CVD.

Supplemental Information

Supplemental Information 1 Raw data.

Click here for additional data file.

Additional Information and Declarations

Competing Interests

Author Contributions

Human Ethics

Data Availability

The authors declare that they have no competing interests.

Yuan Lin conceived and designed the experiments, performed the experiments, authored or reviewed drafts of the article, and approved the final draft.

Fang Zhou performed the experiments, analyzed the data, authored or reviewed drafts of the article, and approved the final draft.

Xihui Wang conceived and designed the experiments, prepared figures and/or tables, authored or reviewed drafts of the article, and approved the final draft.

Yaju Guo performed the experiments, analyzed the data, prepared figures and/or tables, and approved the final draft.

Weiguo Chen analyzed the data, prepared figures and/or tables, authored or reviewed drafts of the article, and approved the final draft.

The following information was supplied relating to ethical approvals (i.e., approving body and any reference numbers):

All samples obtained in this study were approved by the ethics committee of the Second Affiliated Hospital of Xi’an Medical University and abided by the ethical guidelines of the Declaration of Helsinki.

The following information was supplied regarding data availability:

The raw data is available in the Supplemental Files.

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
