# Peer review of "Effect of the index of cardiac electrophysiological balance on major adverse cardiovascular events in patients with diabetes complicated with coronary heart disease"

_PeerJ, doi:10.7717/peerj.15969_

## Round 0.1 · original submission · Minor Revisions

This study investigates the association between ICEB levels and major adverse cardiac events (MACE) in patients with diabetes mellitus complicated with coronary heart disease. The authors aim to determine the risk of MACE based on different ICEB levels and evaluate the effectiveness of a two-stage linear regression model in determining the threshold effect. Overall, this current study provides valuable insights into the relationship between ICEB levels and MACE. However, there are some weaknesses and issues that need to be addressed before publication. Please revise your paper according to the reviewers' opinions, and also in light of my comments.

Here are my comments:
1. Figure 1 needs to be improved. What variables are represented by the horizontal and vertical axes? And legend must also clearly explain what the graph means.
2. [3.2 Outcomes of included patients]: The description of Figure 1 and Table 2 in this section is inadequate. It is difficult for the readers to understand the meaning of the results in this section. More and more details need to be added in this section.
3. This manuscript lacks clarity and organization in several sections. For example, in the Introduction section, the authors mention that individuals with diabetes face an approximately twofold increase in cardiovascular disease (CVD) risk compared to non-diabetic individuals. However, it would be helpful to provide more specific information on the relationship between diabetes and CVD, such as the mechanisms underlying this increased risk and the specific types of CVD that are more prevalent in diabetic patients.
4. Consistent terminology should be used throughout the manuscript to avoid confusion. For instance, the authors refer to the "cardiac electrophysiological balance index" (ICEB) in the title and abstract, but later refer to it as the "index of cardiac electrophysiological balance".
5. The manuscript includes several abbreviations without providing a clear explanation for each one. For example, ICEB is mentioned without a clear explanation of what it represents and how it is calculated. Providing a brief explanation or definition of each abbreviation upon first use would enhance the clarity of the manuscript.
6. There are grammatical issues in this manuscript. For example, in Table 2, the heading "Major adverse cardiac events in participants" should be revised to "Major Adverse Cardiovascular Events in Participants" for consistency and clarity. Additionally, sentence structures should be revised to improve readability and comprehension.
7. [with a view to enabling prompt detection and intervention by forecasting the corresponding risks in their initial stages]: could be changed to [in order to enable early diagnosis and management by anticipating the associated risks in their early phases].
8. [who annually examined]: should be [who were annually examined].
9. [The Table 3 .... for male, female]: should be [Table 3…. Males, females].

Reviewer 1 ·

Basic reporting

The manuscript effectively presents relevant results aligned with the study's hypotheses without unnecessary digressions. The findings directly address the research questions outlined in the introduction.

Experimental design

Methods described with sufficient detail and information to replicate: The manuscript provides sufficient detail and information regarding the research methods. The study design, data collection procedures, statistical analyses, and any other relevant aspects are described comprehensively, enabling other researchers to replicate the study if desired.

Validity of the findings

The research design and data analysis methods employed in the study are appropriate and satisfy the requirements for producing reliable and valid results.

Additional comments

1. Provide recommendations for future research based on the study's findings and limitations.
2. Discuss potential sources of bias and suggest ways to address them in future research.
3. Clearly articulate the limitations of the research design, methodology, and sample size, and their potential impact on the results.
4. Provide a detailed description of the data collection process, including how variables were measured and any data preprocessing steps. Clearly outline the specific methods used for data collection, including instruments, procedures, and any modifications made. Describe how variables were operationalized and any steps taken to ensure the validity and reliability of the measurements.
5. Expand on why the current approach to cardiovascular disease management is considered inadequate.
6. Provide context or additional explanation for the significance of the two novel ECG markers (ICEB and TDR).
7. Provide a brief explanation of how ICEB is calculated and its physiological relevance.

Reviewer 2 ·

Basic reporting

Yes

Experimental design

Yes

Validity of the findings

Yes

Additional comments

1) Begin the introduction with a clear and concise statement about the purpose of the research. For example, the purpose of this research is to investigate the potential associations between ICEB and major adverse cardiovascular events related to CHD, aiming to enable early detection and intervention by forecasting the corresponding risks in their initial stages.
2) Provide a brief overview of the research methodology and design in the introduction. For example, we employed a case-control study design to examine the relationship between ICEB and major adverse cardiovascular events in individuals with CHD.
3) Provide details about the statistical tests used for intra-group comparisons and inter-group comparisons. For example, Intra-group comparisons were conducted using repeated measures analysis of variance (ANOVA), and inter-group comparisons were performed using paired t-tests. Categorical data comparisons were performed using Chi-square tests.
4) Provide the definition and explanation of the hazard ratio (HR) and its interpretation.
5) Describe any potential limitations or confounding factors that may have influenced the results, such as other co-existing medical conditions, medication use, or follow-up duration. This will provide a more comprehensive understanding of the study's findings.
6) Double-check the text for any grammatical errors or unclear phrases. Ensure that the information is presented in a concise and easily understandable manner.

Reviewer 3 ·

Basic reporting

a) In introduction, provide examples or references supporting the potential association between ICEB and major adverse cardiovascular events.
b) The introduction should Include a statement about the potential significance or novelty of the research.

Experimental design

c) Clarify the timeframe for data collection and participant inclusion.
d) Specify any exclusion criteria applied when selecting participants.
e) Specify the number of participants in each ICEB group (ICEB 1, ICEB 2, ICEB 3) for a better understanding of the distribution.

Validity of the findings

f) The manuscript lacks an assessment of the impact and novelty of the findings. Evaluating the potential impact on the field and the novelty of the study's findings would enhance the validity assessment..
g) Discuss the clinical significance of the study findings and their potential implications for patient management or future research. This will help readers understand the importance of the results and their potential impact in the field.

Additional comments

None

---

## Round 0.2 · accepted · Accept

In carefully evaluating the content of this revised paper, I was satisfied with the responses and revisions made by the authors. The Reviewer's and my concerns have been well addressed. With the necessary revisions and improvements, the quality of this paper has been significantly improved. I believe that this revised manuscript is ready to be considered for publication in this journal.